# Ultra-Rapid serial visual presentation reveals dynamics of feedforward and feedback processes in the ventral visual pathway

Yalda Mohsenzadeh[1], Sheng Qin[1], Radoslaw M Cichy[2], Dimitrios Pantazis[1]*

[1]McGovern Institute for Brain Research, Massachusetts Institute of Technology, Cambridge, United States; [2]Education and Psychology, Freie Universität Berlin, Berlin, Germany

**Abstract** Human visual recognition activates a dense network of overlapping feedforward and recurrent neuronal processes, making it hard to disentangle processing in the feedforward from the feedback direction. Here, we used ultra-rapid serial visual presentation to suppress sustained activity that blurs the boundaries of processing steps, enabling us to resolve two distinct stages of processing with MEG multivariate pattern classification. The first processing stage was the rapid activation cascade of the bottom-up sweep, which terminated early as visual stimuli were presented at progressively faster rates. The second stage was the emergence of categorical information with peak latency that shifted later in time with progressively faster stimulus presentations, indexing time-consuming recurrent processing. Using MEG-fMRI fusion with representational similarity, we localized recurrent signals in early visual cortex. Together, our findings segregated an initial bottom-up sweep from subsequent feedback processing, and revealed the neural signature of increased recurrent processing demands for challenging viewing conditions.

DOI: https://doi.org/10.7554/eLife.36329.001

**\*For correspondence:**
pantazis@mit.edu

**Competing interests:** The authors declare that no competing interests exist.

## Introduction

The human visual system interprets the external world through a cascade of visual processes that overlap in space and time. Visual information is transformed not only feedforward, as it propagates through ascending connections, but also from higher to lower hierarchy areas through descending feedback connections and within the same areas through lateral connections (*Ahissar et al., 2009*; *Bullier, 2001*; *Enns and Di Lollo V, 2000*; *Lamme and Roelfsema, 2000*; *Lamme et al., 1998*). This concurrent activation of a dense network of anatomical connections poses a critical obstacle to the reliable measurement of recurrent signals and their segregation from feedforward activity. As a result, our knowledge on the role of recurrent processes and how they interact with feedforward processes to solve visual recognition is still incomplete.

Here we used an ultra-rapid serial visual presentation (ultra-RSVP) of real-world images to segregate early bottom-up from recurrent signals in the ventral pathway. We postulated that, under such rapid stimulus presentations, visual processes will degrade substantially by suppressing sustained neural signals that typically last hundreds of milliseconds. As a result, neural signals would become transient, reducing the overlap of processes in space and time and enabling us to disentangle distinct processing steps. Recent behavioral evidence exemplified the remarkable robustness of the human visual system to capture conceptual information in stimuli presented at similar rates

**eLife digest** The human brain can interpret the visual world in less than the blink of an eye. Specialized brain regions process different aspects of visual objects. These regions form a hierarchy. Areas at the base of the hierarchy process simple features such as lines and angles. They then pass this information onto areas above them, which process more complex features, such as shapes. Eventually the area at the top of the hierarchy identifies the object. But information does not only flow from the bottom of the hierarchy to the top. It also flows from top to bottom. The latter is referred to as feedback activity, but its exact role remains unclear.

Mohsenzadeh et al. used two types of imaging to map brain activity in space and time in healthy volunteers performing a visual task. The volunteers had to decide whether a series of images that flashed up briefly on a screen included a face or not. The results showed that the brain adapts its visual processing strategy to suit the viewing conditions. They also revealed three key principles for how the brain recognizes visual objects.

First, if early visual information is incomplete – for example, because the images appeared only briefly – higher regions of the hierarchy spend more time processing the images. Second, when visual information is incomplete, higher regions of the hierarchy send more feedback down to lower regions. This leads to delays in identifying the object. And third, lower regions in the hierarchy – known collectively as early visual cortex – process the feedback signals. This processing takes place at the same time as the higher levels identify the object.

Knowing the role of feedback is critical to understanding how the visual system works. The next step is to develop computer models of visual processing. The current findings on the role of feedback should prove useful in designing such models. These might ultimately pave the way to developing treatments for visual impairments caused by damage to visual areas of the brain.

DOI: https://doi.org/10.7554/eLife.36329.002

(*Broers et al., 2018*; *Evans et al., 2011*; *Potter et al., 2014*). Thus the underlying neural signals, while deprived, would still represent brain activity required to accomplish visual object recognition.

We recorded human MEG data while participants viewed ultra-RSVP sequences with rates 17 or 34 ms per picture. Confirming our hypothesis, the rapid presentation of images segregated the activation cascade of the ventral visual pathway into two temporally dissociable processing stages, disentangling the initial bottom-up sweep from subsequent processing in high-level visual cortex. Capitalizing on this dissociation, we used multivariate pattern classification of MEG data to characterize the activation dynamics of the ventral pathway and address the following three challenges: we investigated how the evolution of the bottom-up sweep predicts the formation of high-level visual representations; we sought evidence for rapid recurrent activity that facilitates visual recognition; and we explored whether reducing visibility with higher stimulus presentation rates increases recurrent processing demands. Finally, to resolve the locus of feedforward and feedback visual signals, we tracked the spatiotemporal dynamics with a MEG-fMRI fusion approach based on representational similarity (*Cichy et al., 2014*, *2016a*; *Kriegeskorte et al., 2008*).

## Results

We collected MEG data while human participants (n = 17) viewed rapid sequences of 11 real-world images presented at 17 or 34 ms per picture. The middle image (sixth image, named target) was randomly sampled from a set of 12 face images or 12 object images, while the remaining images (1–5 and 7–11, named masks) comprised different categories of objects (*Figure 1a*). Participants performed a two-alternative forced choice task reporting whether a face was present in the sequence or not.

Even though the image presentation was extremely rapid, participants performed the face detection task consistently above chance in the 17 ms per picture RSVP condition (sensitivity index $d' \pm$ SEM = $1.95 \pm 0.11$), and with high accuracy in the 34 ms per picture RSVP condition ($d' \pm$ SEM = $3.58 \pm 0.16$) (*Figure 1b*). Behavioral performance was significantly different between the two conditions ($n=17$; two-sided signed-rank test; $P \ll 0.001$).

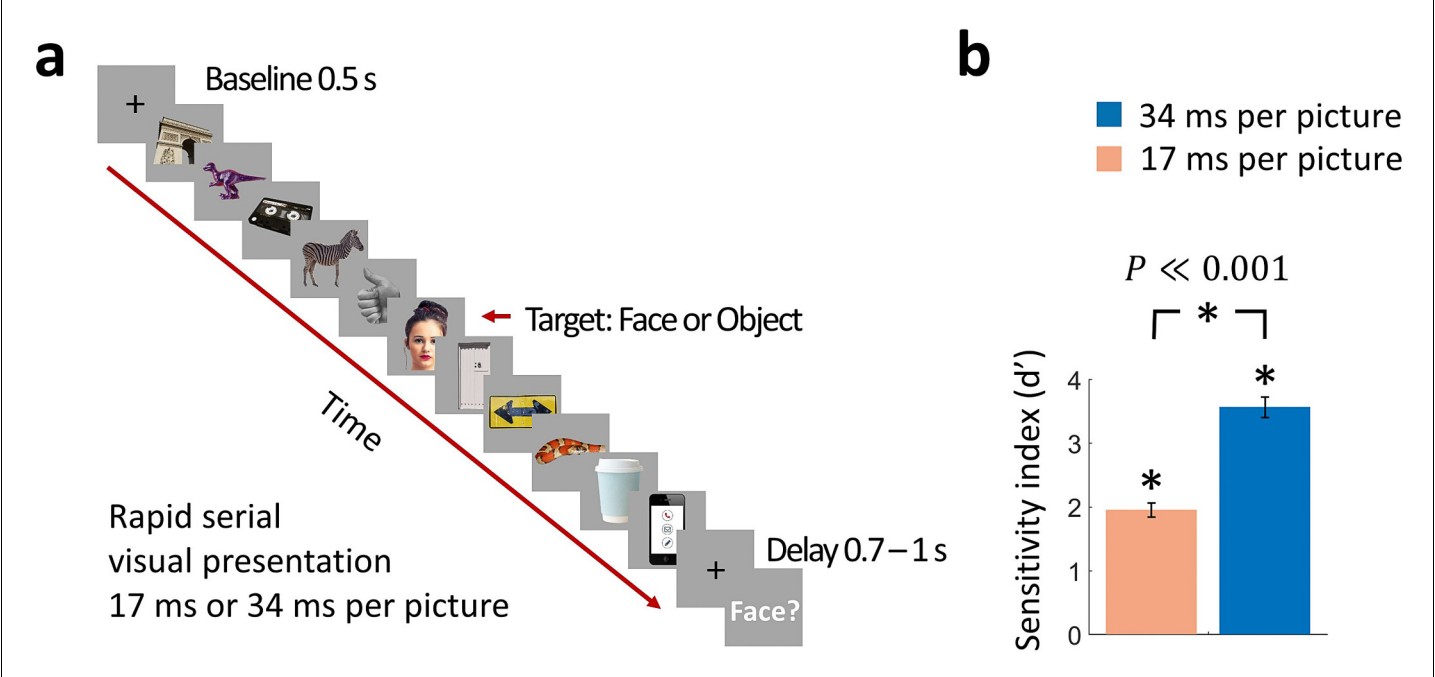

**Figure 1.** Rapid serial visual presentation (RSVP) task. (a) Experimental procedure. The stimulus set comprised 12 face targets, 12 object targets, and 45 masks of various objects. Participants viewed a RSVP sequence of 11 images, with the middle image (target) either a face or an object. The images were presented at a rate of 17 ms per picture or 34 ms per picture in separate trials. Following a delay of 0.7 - 1 s to prevent motor artifacts, subjects were prompted to respond by pressing a button whether they have seen a face or not. (Images shown are not examples of the original stimulus set due to copyright; the exact stimulus set is visualized at https://megrsvp.github.io. Images shown are in public domain and available at pexels.com under a Creative Commons Zero (CC0) license.) (b) Behavioral performance in the two RSVP conditions. Bars indicate d' performance and error bars indicate SEM. Stars above each bar and between bars indicate significant performance and significant differences between the two conditions, respectively (n=17; two-sided signed-rank test; P≪0.001).

DOI: https://doi.org/10.7554/eLife.36329.003

To track how neural representations resolved stimulus information in time, we used time-resolved multivariate pattern classification on the MEG data (*Cichy and Pantazis, 2017*; *Cichy et al., 2014*; *Isik et al., 2014*; *King and Dehaene, 2014*). We extracted peri-stimulus MEG signals from −300 ms to 900 ms (1 ms resolution) with respect to the target image onset. For each time point separately, we used the MEG data to classify pairwise (50% chance level) all 24 target images. The results of the classification (% decoding accuracy) were used to populate a time-resolved 24 × 24 decoding matrix indexed by the target images (*Figure 2a*). To demonstrate the advantage of RSVP in dissociating visual processes against other experimental paradigms, we further computed decoding matrices in a slow visual presentation at 500 ms per picture, where the same 24 target images were presented in isolation for 500 ms with an ISI of 1 s. The entire procedure yielded three time-resolved decoding matrices, one for the 17, 34, and 500 ms per picture conditions respectively.

## Rapid serial visual presentation disrupted the early sweep of visual activity

To determine the time series with which individual images were discriminated by neural representations, we averaged all elements of the decoding matrix, resulting in a grand total decoding time series (*Figure 2b*). First, we found that neural responses were resolved at the level of individual images in all 3 viewing conditions. Second, decoding accuracies decreased with faster stimulus presentation rates, reflecting the challenging nature of the RSVP task with stimuli presented for very short times. Third, peak latencies shifted earlier with faster stimulus presentation rates (*Figure 2e*). That is, the 500 ms per picture condition reached a peak at 121 ms (95% confidence interval: 102-126 ms), preceded by the 34 ms per picture RSVP condition at 100 ms (94-107 ms), and finally the 17 ms per picture RSVP condition at 96 ms (93-99 ms) (all statistically different; P<0.05; two-sided

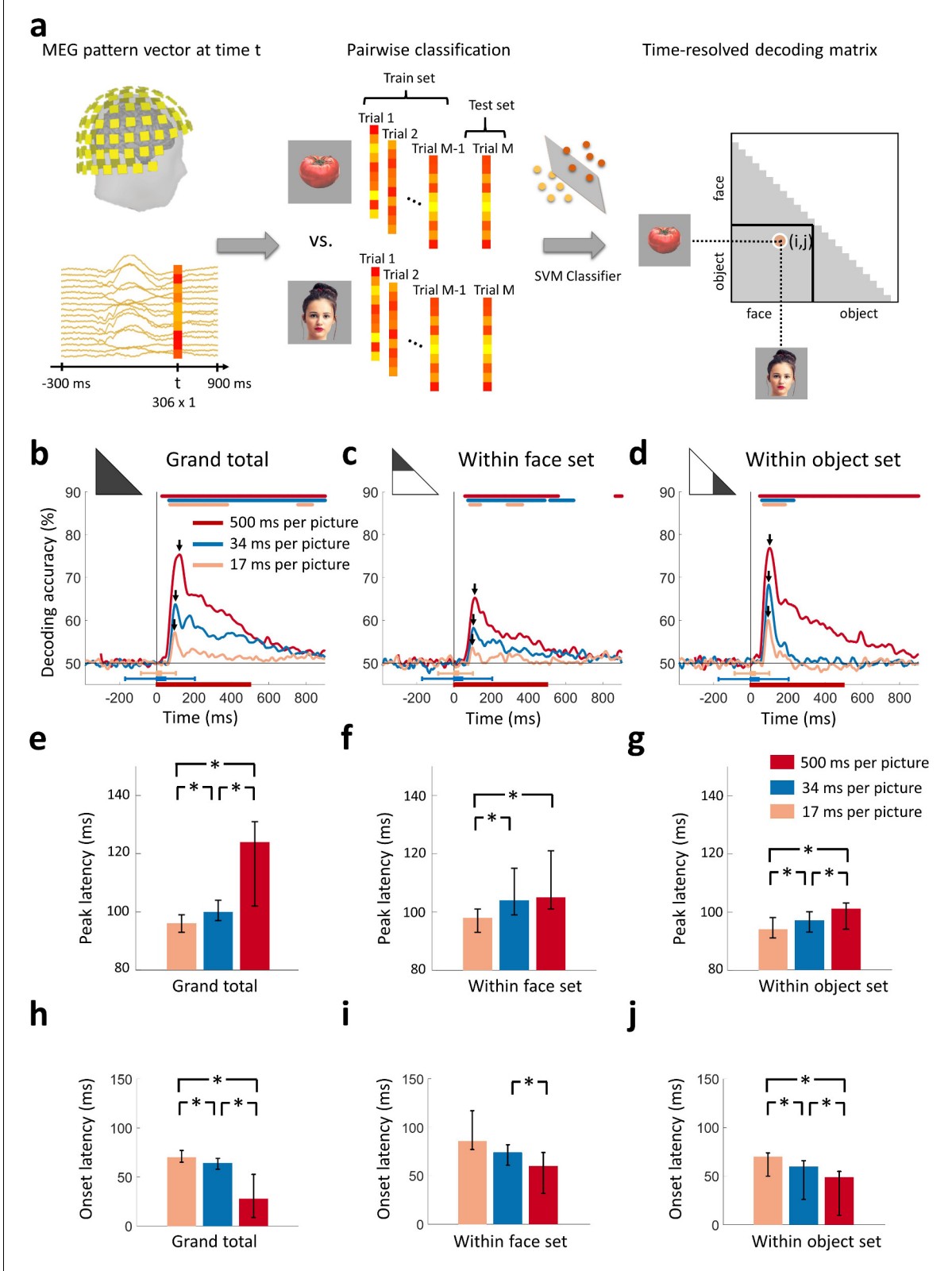

**Figure 2.** Decoding of target images from MEG signals. (**a**) Multivariate pattern analysis of MEG signals. A support vector machine (SVM) classifier learned to discriminate pairs of target images using MEG data at time point *t*. The decoding accuracies populated a 24 × 24 decoding matrix at each time point *t*. (Images shown are not examples of the original stimulus set due to copyright; see *Figure 1* caption for details) (**b**) Time course of grand total target image decoding for the 500, 34, and 17 ms per picture conditions. Pairwise decoding accuracies were averaged across all elements of the

*Figure 2 continued on next page*

*Figure 2 continued*

decoding matrix. Time is relative to target image onset. Color coded lines above plots indicate significant times. Color coded interval lines below plots indicate stimuli presentation times, with thick and thin lines indicating target and mask presentations. Arrows indicate peak latencies. (c, d) Time course of within category target image decoding for faces and objects. The decoding matrix was divided into 2 segments for pairs of within-face and within-object comparisons, and the corresponding decoding accuracies were averaged. (e–g) Peak latency times for the above target decoding time courses and corresponding 95% confidence intervals are depicted with bar plots and error bars, respectively. (h–j) Onset latency times for the above decoding time courses and 95% confidence intervals. Stars above bars indicate significant differences between conditions. (*n*=16 for 500 ms per picture and *n*=17 for RSVP conditions; time courses were evaluated with one-sided sign permutation tests, cluster defining threshold *P*<0.05, and corrected significance level *P*<0.05; bar plots were evaluated with bootstrap tests for 95% confidence intervals and two-sided hypothesis tests; false discovery rate corrected at *P*<0.05).

DOI: https://doi.org/10.7554/eLife.36329.004

sign permutation tests). Fourth, onset latencies shifted later with faster stimulus presentation rates (*Figure 2h*). That is, the 500 ms per picture condition had onset at 28 ms (9-53 ms), followed by the 34 ms per picture RSVP condition at 64 ms (58-69 ms), and finally the 17 ms per picture RSVP condition at 70 ms (63-76 ms) (all statistically different; *P*<0.05; two-sided sign permutation tests).

The decreased decoding accuracy combined with the increasingly early peak latency and increasingly late onset latency for the RSVP conditions indicate that visual activity was disrupted over the first 100 ms. Even though the highest levels of the visual processing hierarchy in humans are reached with the first 100 ms, there is little time for feedback connections from these areas to exert an effect (*Lamme and Roelfsema, 2000*). Thus, neural activity during the first 100 ms has been linked to the engagement of feedforward and local recurrent connections, rather than long-range feedback connections. In line with these arguments, the early peaks at 100 and 96 ms for the 34 and 17 ms per picture RSVP conditions, respectively, explicitly delineate the first sweep of visual activity, differentiating it from later neural activity that includes feedback influences from the top of the visual hierarchy. Further, if early decoding would only reflect feedforward activity, we would not expect to see onset latency differences, but we do. The fact that different stimulus durations have different onsets suggests that interactions with recurrent activity are already incorporated when the first decoding onsets emerge, arguing against the view that the early part of the decoding time course can be uniquely tied to feedforward alone (*Fahrenfort et al., 2012*; *Lamme and Roelfsema, 2000*; *Ringach et al., 1997*).

Next, to investigate the generalization of our findings to any pair of images, even when they share categorical content, we evaluated whether our results hold to within-category image classification. For this, we subdivided the decoding matrix into two partitions, corresponding to within-face comparisons, and within-object comparisons. Averaging the elements of each partition separately determined the time series with which individual images were resolved within the subdivision of faces (*Figure 2b*) and objects (*Figure 2c*). We confirmed the generalization and reliability of our findings, as our results were similar to the grand total decoding time series: individual images were discriminated by neural responses; decoding accuracies were weaker for rapid stimulus presentations; and peak and onset latencies had the same ordinal relationship as in the grand total analysis (*Figure 2fg*). Peak and onset latencies for the grand total and within category comparisons are shown in *Table 1*.

In sum, decoding accuracies decreased with progressively shorter stimulus presentation times, indicating that neuronal signals encoded less stimulus information at rapid presentation rates. Onset latencies shifted late with shorter presentation times, indicating that recurrent activity exerts its influence even as the first decoding onsets emerged. Importantly, the progressively earlier peak with shorter presentation times indicated disruption of the first sweep of visual activity, thus indexing feedforward and local recurrent processing and segregating it in time from subsequent processing that includes feedback influences from high-level visual cortex.

## Rapid serial visual presentation delayed the emergence of categorical information

How did the disruption of the early sweep of visual activity, reported in the previous section, affect the emergence of categorical information in the RSVP conditions? A prevalent theory posits that core object recognition is largely solved in a feedforward manner (*DiCarlo et al., 2012*; *Liu et al.,*

**Table 1.** Peak and onset latency of the time series for single image decoding (**Figure 2**) and categorical division decoding (**Figure 3**), with 95% confidence intervals in brackets.

| | Presentation rate | Peak latency (ms) | Onset latency (ms) |
|---|---|---|---|
| Grand total | 500 ms per picture | 121 (102–126) | 28 (9–53) |
| | 34 ms per picture | 100 (96–107) | 64 (58–69) |
| | 17 ms per picture | 96 (93–99) | 70 (63–76) |
| Within-faces | 500 ms per picture | 113 (104–119) | 59 (30–73) |
| | 34 ms per picture | 104 (96–109) | 74 (62–81) |
| | 17 ms per picture | 98 (96–104) | 86 (78–117) |
| Within-objects | 500 ms per picture | 102 (93–102) | 48 (10–55) |
| | 34 ms per picture | 97 (90–97) | 60 (27–67) |
| | 17 ms per picture | 94 (87–95) | 70 (64–74) |
| Between minus within | 500 ms per picture | 136 (130–139) | 46 (15–51) |
| | 34 ms per picture | 169 (165–177) | 73 (67–78) |
| | 17 ms per picture | 197 (175–218) | 139 (67–155) |

DOI: https://doi.org/10.7554/eLife.36329.005

*2002*; *Serre et al., 2007*; *Thorpe et al., 1996*). If this holds under rapid presentation conditions, then categorical signals would be expected to emerge with comparable dynamics regardless of stimulus presentation rates. However, opposing theories concur that feedback activity is critical for visual awareness and consciousness (*Lamme and Roelfsema, 2000*; *Ahissar et al., 2009*; *Fahrenfort et al., 2017*, *2012*). According to these theories, presenting stimuli at rapid presentation rates would (i) afford less time for initial stimulus evidence accumulation (a process that in all likelihood already incorporates some local recurrent processing, as suggested by variable onset latencies reported in the previous section) and (ii) lead to disruption of recurrent signals of the target stimulus due to the masking stimuli of the RSVP paradigm. These would be consistent with slowing down the speed and extent with which category information can be resolved using recurrence (*Brincat and Connor, 2006*; *Tang and Kreiman, 2017*).

To differentiate between those competing theories, we computed categorical division time series. We divided the decoding matrix into partitions corresponding to within-category (face or object) and between-category stimulus comparisons separately for each of the three viewing conditions (*Figure 3a*). The difference of between-category minus within-category average decoding accuracies served as a measure of clustering by category membership.

We found that the categorical division time series resolved face versus object information in all three conditions (*Figure 3a*). Consistent with the grand total decoding results, categorical neural representations were stronger in the 500 and 34 ms per picture conditions than the 17 ms per picture condition. Multidimensional scaling (MDS) plots (*Kruskal and Wish, 1978*) at peak latencies for the three conditions, offering an intuitive visualization of the stimulus relationships, are shown in *Figure 3c*. These plots revealed strong categorical division for the 500 and 34 ms per picture conditions, followed by weaker but still distinct categorical division in the 17 ms per picture condition.

The peak of the categorical division time series revealed the time points at which categorical information was most explicitly encoded in the neural representations (*DiCarlo and Cox, 2007*). The peak latency increased as presentation rates became progressively faster. That is, the time series for the 500 ms per picture condition peaked at 136 ms (130-139 ms), followed by the 34 ms per picture RSVP at 169 ms (165-177 ms), and the 17 ms per picture RSVP at 197 ms (184-218 ms) (all statistically different; $P<0.05$; two-sided sign permutation tests) (*Figure 3b*). This relationship is reverse from the peak latency of the first sweep of visual activity reported in the previous section, further stressing the existence of variable dynamics in the ventral pathway. This suggests that categorical information did not arise directly in a purely feedforward mode of processing, as this would predict comparable temporal dynamics in all conditions. Instead, it is consistent with the idea that recurrent interactions within the ventral stream facilitate the emergence of categorical information by enhancing stimulus

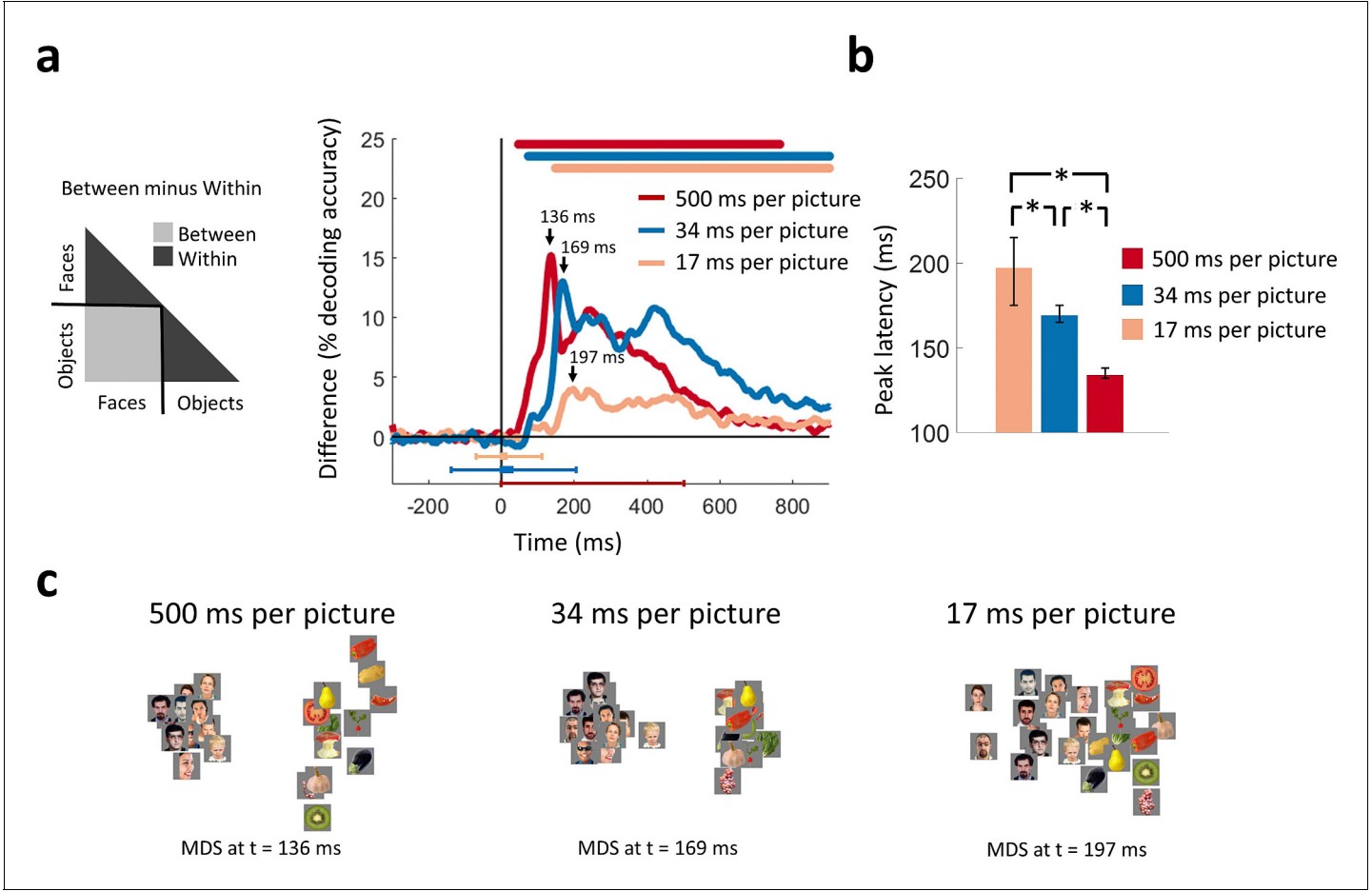

**Figure 3.** Categorical information encoded in MEG signals. (**a**) Time course of categorical division depending on presentation rate. For each condition, the MEG decoding matrix was divided into 3 segments for pairs of within-face, within-object, and between-face/object comparisons, and a categorical effect was estimated by contrasting the averaged decoding accuracies of the within from the between segments. Time is relative to target image onset. Color coded lines above plots, interval bars below plots, and arrows same as in *Figure 2*. (**b**) Peak latency times for the categorical information time courses and 95% confidence intervals are depicted with bar plots and error bars, respectively. Stars above bars indicate significant differences between conditions. (**c**) The first two dimensions of multidimensional scaling (MDS) of the MEG decoding matrices are shown for the times of peak categorical information for the 3 conditions. (*n*=16 for 500 ms per picture and *n*=17 for RSVP conditions; time courses were evaluated with one-sided sign permutation tests, cluster defining threshold *P*<0.05, and corrected significance level *P*<0.05; bar plots were evaluated with bootstrap tests for 95% confidence intervals and two-sided hypothesis tests; false discovery rate corrected at *P*<0.05).

DOI: https://doi.org/10.7554/eLife.36329.006

The following figure supplement is available for figure 3:

**Figure supplement 1** Linear decoding of faces vs. objects category.
DOI: https://doi.org/10.7554/eLife.36329.007

information in challenging visual tasks (*Brincat and Connor, 2006*; *Hochstein and Ahissar, 2002*; *Rajaei et al., 2018*; *Tang and Kreiman, 2017*; *Tapia and Beck, 2014*).

Taken together, our results revealed variable temporal neural dynamics for viewing conditions differing in presentation time. Even though the peak latency of the first sweep of visual activity shifted earlier with higher presentation rates, as reported in the previous section, the peak latency of categorical information shifted later, stretching the time between the abrupt end of the initial visual sweep and the emergence of categorical information. This inverse relationship in peak latencies discounts a feedforward cascade as the sole explanation for categorical representations.

## Neuronal representations became increasingly transient at rapid stimulus presentation rates

As neuronal signals propagate along the ventral pathway, neural activity can either change rapidly at subsequent time points, or persist for extended times. Transient activity reflects processing of different stimulus properties over time in either a feedforward manner, as computations become more abstract, or a recurrent manner as neurons tune their responses. On the other hand, persistent activity could maintain results of a particular neural processing stage for later use.

Our premise in introducing the ultra-RSVP task was to suppress the persistent neural activity, and in doing so better capture the transient neural dynamics that reflect distinct neural processing steps. To experimentally confirm that persistent neural activity was indeed suppressed with rapid presentation rates, we extended the SVM classification procedure with a temporal generalization approach (*Cichy et al., 2014*; *Isik et al., 2014*; *King and Dehaene, 2014*). In particular, we used a classifier trained on data at a time point *t* to evaluate discrimination at all other time points *t'*. Intuitively, if neural representations are sustained across time, the classifier should generalize well across other time points.

Temporal generalization matrices were computed by averaging decoding across all pairwise image conditions and all subjects, thus extending over time the results presented in *Figure 2b*. Our temporal generalization analysis confirmed that neural activity became increasingly transient at rapid presentation rates (*Figure 4*). While the 500 ms per picture condition had maps with broad off-diagonal significant elements characteristic of sustained representations, the RSVP conditions had narrow diagonal maps indicating transient neural patterns, with the 17 ms per picture RSVP narrower than the 34 ms per picture RSVP.

The increasingly transient activity in the RSVP conditions shows that neural activity continuously transformed stimulus information in a feedforward and feedback manner, will less neural resources used to maintain information. Thus, the results confirmed our hypothesis that the ultra-RSVP task would suppress persistent neural activity.

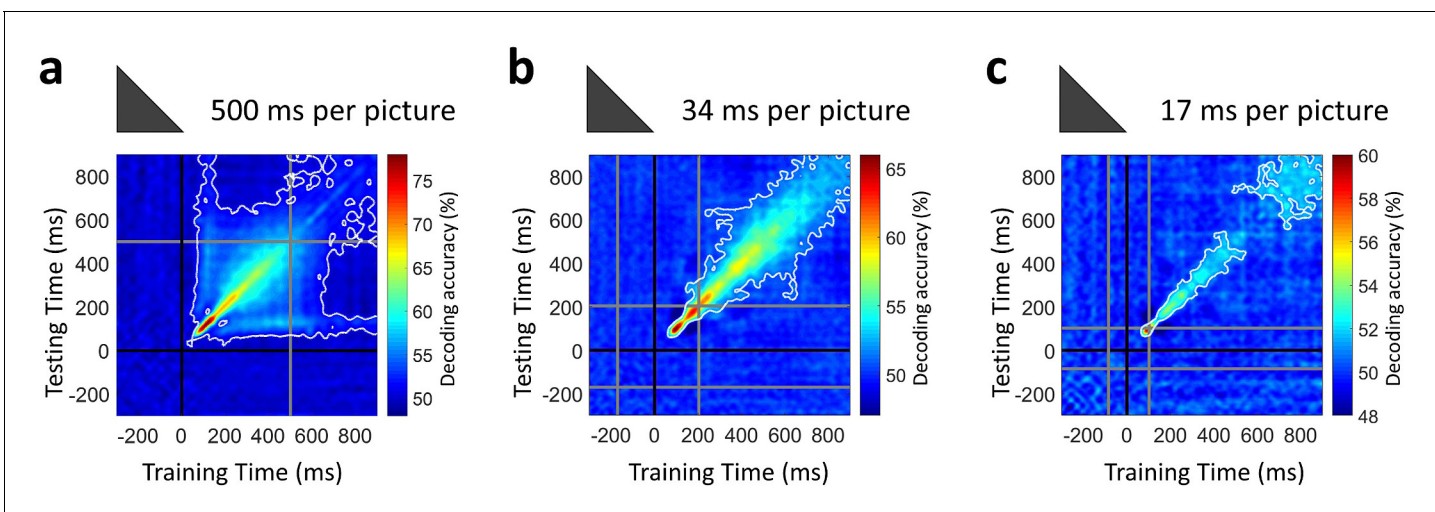

**Figure 4.** Temporal generalization of target image decoding for the 500, 34, and 17 ms per picture conditions. The SVM classifier was trained with MEG data from a given time point *t* (training time) and tested on all other time points (testing time). The temporal generalization decoding matrix was averaged over all image pairs and all subjects, thus corresponding to the temporal generalization of the grand total decoding time series in *Figure 2b*. The black line marks the target image onset time. The gray lines mark the image offset in the 500 ms per picture condition and the RSVP sequence onset/offset times in the rapid presentation conditions. The white contour indicates significant decoding values (*n*=16 for 500 ms per picture and *n*=17 for RSVP conditions; one-sided sign permutation tests, cluster defining threshold *P*<0.05, and corrected significance level *P*<0.05).
DOI: https://doi.org/10.7554/eLife.36329.008

## Unfolding the dynamics of feedforward and feedback processes in space and time

The analyses presented thus far segregated the temporal dynamics of the initial bottom-up sweep from subsequent signals incorporating recurrent activity in the ventral pathway. Furthermore, peak latencies for early and late visual signals varied inversely, consistent with feedback processing. Here we mapped visual signals on the cortex to identify where in the brain feedforward and feedback signal interact.

To map the spatiotemporal dynamics of the visual processes we used a MEG-fMRI fusion method based on representational similarity (*Cichy et al., 2014*, *2016a*). For this, we first localized the MEG signals on the cortex and derived the time series from all source elements within two regions-of-interest (ROIs): early visual cortex (EVC) and inferior temporal cortex (IT). We selected EVC as the first region of the cortical feedforward sweep, and IT as the end point where neural patterns have been found to indicate object category (*Cichy et al., 2014*). We then performed time-resolved multivariate pattern classification on the MEG data following the same procedure described earlier, only now we created pattern vectors by concatenating activation values from sources within a given ROI, instead of concatenating the whole-head sensor measurements. This procedure resulted in one MEG RDM for each ROI and time point.

We compared the representational similarity between the time-resolved MEG RDMs for the two cortical regions (EVC and IT) and the fMRI RDMs for the same regions (*Figure a and b*). This yielded two time series of MEG-fMRI representational similarity, one for EVC and one for IT. In all conditions, consistent with the view of visual processing as a spatiotemporal cascade (*Cichy et al., 2014*), the time series peaked earlier for EVC than IT (*Figure 5c–h*). The peak-to-peak latency between EVC and IT increased as viewing conditions became increasingly challenging with faster presentation rates: $\Delta$=27 ms for the 500 ms per picture condition; $\Delta$=79 ms for the 34 ms per picture RSVP; and $\Delta$=115 ms for the 17 ms per picture RSVP (all statistically different; two-sided bootstrap hypothesis tests; $P$<0.05). This latency difference was the compounded effect of two factors. First, the EVC peak had progressively shorter latencies (104 vs. 87 vs. 80 ms for the 3 conditions), and second the IT peak had progressively longer latencies (131 vs. 166 vs. 195 ms for the three conditions). This inverse relationship between the EVC and IT peaks corroborated the findings of the previous sections, namely that a disrupted first sweep of visual activity was associated with a delayed emergence of categorical division information. It further bound the processing stages in time to the V1 and IT locations in space.

Importantly, while EVC had a single peak at 104 ms and persistent representations over hundreds of milliseconds for the 500 ms per picture condition, its dynamics were transient and bimodal for the RSVP conditions. For the 34 ms per picture RSVP condition, an early peak at 87 ms was immediately followed by weak MEG-fMRI representational similarities, and then a second peak at 169 ms. For the 17 ms per picture RSVP condition, we observed similar dynamics with an early peak at 80 ms and a second peak at 202 ms, though in this case the second peak was not strictly defined because the time course did not reach significance, possibly due to compromised neural representations at such fast stimulus presentation rates. The second peak in EVC occurred at similar times as the peak in IT ($\Delta$=3 ms for the 34 ms per picture RSVP condition, $p$=0.06; and $\Delta$=7 ms for the 17 ms per picture RSVP condition, $p\ll$0.001; two-sided bootstrap hypothesis tests). This is consistent with feedback activity in EVC at the same time as IT solves visual object recognition. *Table 2* summarizes latencies and 95% confidence intervals for all conditions. We note here that resolving feedback activity in EVC was possible with MEG-fMRI fusion because MEG activation patterns disentangled slow fMRI hemodynamic responses in EVC that correspond to the combined contributions of feedforward and feedback visual activity.

In sum, the combination of the RSVP paradigm with MEG-fMRI representational similarity resolved bimodal dynamics for EVC. The first EVC peak offered evidence that disruption of early visual activity resulted in delayed categorical division information in IT. The second EVC peak occurred at approximately the same time as the peak in IT and is consistent with feedback activity from IT to EVC.

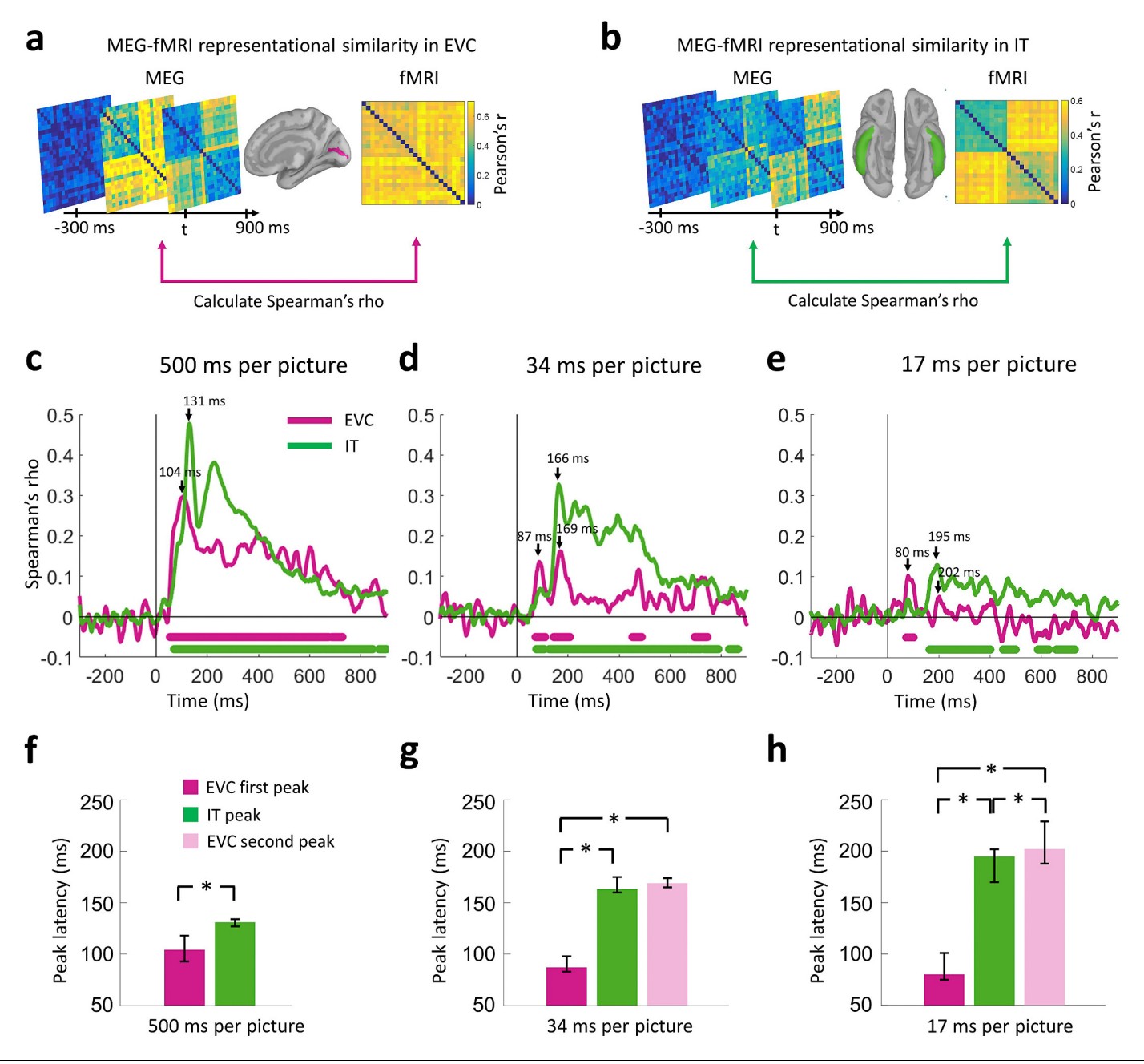

**Figure 5.** Representational similarity of MEG to fMRI signals at EVC and IT. (**a**) For every time point *t*, the EVC-specific MEG RDM was compared (Spearman's rho) with the EVC-specific fMRI RDM, yielding a time series of MEG-fMRI representational similarity at EVC. (**b**) Same as in (**a**) but for IT. (**c–e**) Time series of MEG-fMRI representational similarity at EVC and IT for the 3 conditions. Time is shown relative to target image onset. Color coded lines below plots indicate significant times. Peak latencies are indicated with arrows. While the 202 ms peak of the EVC time series in (**e**) is not significant, it is marked to indicate comparable temporal dynamics with the 34 ms per picture condition. (**f–h**) Peak latency times for the representational similarity time courses and 95% confidence intervals are depicted with bar plots and error bars, respectively. Stars above bars indicate significant differences between conditions. (*n*=16 for 500 ms per picture and *n*=17 for RSVP conditions; time courses were evaluated with one-sided sign permutation tests, cluster defining threshold *P*<0.05, and corrected significance level *P*<0.05; bar plots were evaluated with bootstrap tests for 95% confidence intervals and two-sided hypothesis tests; false discovery rate corrected at *P*<0.05).
DOI: https://doi.org/10.7554/eLife.36329.010

**Table 2.** Peak and onset latency of the time series for MEG-fMRI fusion at EVC and IT, with 95% confidence intervals in brackets.

|  |  | Peak latency (ms) | Onset latency (ms) |
|---|---|---|---|
| 500 ms per picture | IT | 131 (127–135) | 63 (50–75) |
|  | EVC | 104 (93–120) | 53 (50–57) |
| 34 ms per picture | IT | 166 (162–173) | 70 (50–88) |
|  | EVC | 87 (83–97) and 169 (164-176)* | 71 (61–81) |
| 17 ms per picture | IT | 195 (170–203) | 162 (55–183) |
|  | EVC | 80 (75–100) and 202 (190-219)* | 71 (50–76) |

*Time series had two early peaks.
DOI: https://doi.org/10.7554/eLife.36329.009

## Discussion

Using an ultra-RSVP task, we dissected the ventral pathway activation cascade into two temporally distinct processing stages: the initial bottom-up sweep and the subsequent emergence of categorical information. For the first stage, we found a progressively earlier peak with decreasing viewing times, indicating an early disruption of visual activity (*Figure 2*). For the second stage, we found a progressively later peak with decreasing viewing times, indicating a delayed emergence of categorical information at high-level visual cortex (*Figure 3*). This reverse relation of the peak latencies between the two processing stages has two critical implications: first, the extent of disruption of the initial sweep is related to longer processing times at subsequent stages to solve visual recognition; and second, such variable temporal dynamics index the existence of recurrent activity in the ventral pathway that takes up additional processing time to solve visual recognition when initial signals are limited in time. Finally, using MEG-fMRI fusion we pinpointed the locus where recurrent activity becomes effective to EVC (*Figure 5*), with temporal onset overlapping with the dynamics of the emergence of categorical information in high-level cortex.

While a large body of literature has investigated recurrent processing in the visual cortex, our work goes beyond and complements prior studies in several dimensions. First, consistent with behavioral and transcranial magnetic stimulation experiments that postulated the locus of recurrent activity signals in V1/V2 (*Camprodon et al., 2010*; *Drewes et al., 2016*; *Koivisto et al., 2011*; *Wokke et al., 2013*), we mapped the locus of rapid feedback activity in early visual cortex. Second, our study elucidates the functional nature of recurrent connections in vision. Recurrent connections have often been thought to subserve slower top-down attentional modulations from frontoparietal sites once recognition has been solved (*Bar et al., 2006*; *Hopf et al., 2006*; *Sehatpour et al., 2008*). In contrast, our study corroborates growing experimental evidence suggesting that recurrent signals also originate from within the ventral pathway as recognition unfolds, and can exert their influence before attentional modulations (*Boehler et al., 2008*; *Wyatte et al., 2014*). Third, while extensive evidence suggests the existence of rapid feedback signals using highly controlled artificial stimuli, such as Kanizsa-type illusory figures, motion of random dots, and oriented bars (*Halgren et al., 2003*; *Hupé et al., 1998*; *Lamme et al., 2002*; *Murray et al., 2002*; *Ringach et al., 1997*; *Wokke et al., 2012*, *2013*; *Yoshino et al., 2006*), it is not clear whether these results generalize to real-world situations. Since our target stimuli were natural images of objects and faces, we demonstrated the existence of feedback activity in early visual cortex under more ecologically valid conditions. Fourth, even though temporal delays in processing stimulus information have been observed when visibility conditions were impoverished (*Tang and Kreiman, 2017*), here we explicitly related bottom-up and feedback dynamics to each other and offered well-defined temporal signatures for both mechanisms. Last, rather than using unimodal data to study recurrent activity that reveals either spatial or temporal aspects of brain activity, we applied a novel way to investigate ventral neural dynamics by coupling MEG and fMRI data to holistically capture the spatiotemporal dynamics of brain activation (*Cichy et al., 2014*, *2016a*).

Our results thus allow the identification of three fundamental processing principles of object recognition: (i) the brain compensates for early disruptions of the initial sweep through longer processing times at subsequent stages; (ii) delays in solving object recognition are an index of increased recurrent processing; and (iii) such recurrent processing takes place in early visual cortex and coincides in time with the emergence of categorical information in inferior-temporal cortex.

## Sequencing the processing cascade in the ventral visual pathway

Here we discuss the nature of visual processing and the role of recurrent dynamics for each of the two temporally distinct stages of visual processing revealed by the ultra-RSVP task.

### First stage - feedforward activity

The first wave of visual responses we observed peaked at approximately 100 ms for the RSVP conditions (*Figure 2b*). This latency falls within the time range during which the feedforward sweep is expected to reach the top of the visual hierarchy, but leaves little time for substantial contributions from high-level feedback connections (*Lamme and Roelfsema, 2000*; *Tapia and Beck, 2014*; *Wyatte et al., 2014*). Therefore, responses up to 100 ms tracked a feedforward process with contributions from local recurrent circuits, but separate from later feedback signals emerging from high-level visual cortex.

### First stage - early recurrent activity

The first sweep of visual activity was characterized by a ramping process with a peak latency approximately 100 ms for the RSVP conditions (*Figure 2b*). Such extended latencies, though too short to include substantial feedback signals from the top of the visual hierarchy, permit incorporation of information from local recurrent connections. In typical viewing conditions, such as the 500 ms per picture condition used here, neurons are known to remain active even after their initial contribution in the feedforward sweep, with their responses modulated with contextual information both within and outside their receptive fields (*Lamme and Roelfsema, 2000*). Examples include the dynamic orientation tuning of V1 cell responses, which is shaped by horizontal intracortical connections (*Ringach et al., 1997*), and the segregation of texture, which is shaped by feedback signals from extrastriate areas (*Hupé et al., 1998*; *Lamme et al., 1998*). Experiments using transcranial magnetic stimulation (*Camprodon et al., 2010*; *Koivisto et al., 2011*), backward masking (*Boehler et al., 2008*; *Fahrenfort et al., 2007*; *Kovács et al., 1995*; *Lamme et al., 2002*), and reversible cooling (*Hupé et al., 1998*) offer evidence that rapid feedback circuits in the ventral pathway are engaged within the first 80–150 ms of vision, with local feedback signals from extrastriate areas to V1 emerging within 80–110 ms (*Wyatte et al., 2014*).

Was the first wave in the ultra-RSVP conditions shaped by local recurrent connections? While our results cannot discount such influences, it is reasonable to construe the first wave in the RSVP conditions as disproportionately characterized by feedforward propagating signals rather than local recurrent activity. This is because EVC time series had strong MEG-fMRI representational similarities over hundreds of milliseconds for the 500 ms per picture condition, whereas weak dynamics with early peak latencies (80 and 87 ms) for the RSVP conditions (*Figure 5c–e*). This suggests any possible recurrent processes in EVC for the 500 ms per picture condition were suppressed in the RSVP conditions immediately after 80 or 87 ms.

Reduced early recurrent activity in the RSVP conditions could be due to two reasons. First, shorter stimulus presentation times resulted in limited stimulus evidence accumulation. This may be reflected in the progressively later onset latencies with decreasing viewing times in *Figure 2h–j*, which suggest that local recurrent interactions have a rapid influence in the ventral stream dynamics. And second, the mask stimuli disrupted recurrent processing of the target in the RSVP conditions. This may be reflected in the more transient dynamics for the RSVP conditions in *Figure 4*.

### Second stage - delayed emergence of categorical information

The first sweep of visual activity was temporally dissociated from a second processing stage at which categorical information emerged. The peak latency and strength of the early visual sweep were inversely related with the timing of categorical representations. Thus, compromised visual signals early in the ventral stream resulted in delays in the emergence of categorical information. This is

consistent with recurrent visual processes requiring additional time to interpret the face target images within the RSVP sequences. Such delays in neural responses have been used as an indicator of recurrent computations in previous studies, including object selective responses in human fusiform gyri for partially visible images (*Tang and Kreiman, 2017*) and the integration of object parts into a coherent representation in macaque posterior IT (*Brincat and Connor, 2006*).

An alternative explanation of the variable peak latencies due to delayed propagation of feedforward signals is unlikely. Instead of observing a graded increase in latencies, the peaks in the feedforward sweep were inversely related to the peaks in categorical information, stretching the time difference between the two processing stage and discounting a purely feedforward explanation.

## Second stage - recurrent activity in EVC

Categorical signals for the two RSVP conditions (*Figure 3a*) were associated with the onset of IT responses and the synchronous reengagement of EVC (*Figure 5c–e*). This reengagement of EVC at the same time categorical information emerged in high-level visual cortex suggests recurrent processes as its basis.

Reengagement of EVC is consistent with theories positing that feedback signals serve to add details in an initially established scene (*Hochstein and Ahissar, 2002*; *Peyrin et al., 2010*; *Tapia and Beck, 2014*). According to these theories, when stimuli are degraded, partial, or otherwise ambiguous, recurrent processes fill-in stimulus information by propagating partial responses back to EVC and activating neurons that would normally respond to the complete stimulus (*Muckli et al., 2015*; *O'Reilly et al., 2013*; *Tang and Kreiman, 2017*; *Wyatte et al., 2014*). Thus, here recurrent activity may have filled-in missing visual information necessary to recognize the face target images. Such information is probably behaviorally relevant, as even relatively modest shifts in the latency of categorical information encoded rapidly in the ventral stream have been linked to corresponding shifts in behavioral response delays (*Cauchoix et al., 2016*). Future work could benefit from our neuroimaging methodology to investigate the precise role of recurrent activity in EVC.

While the recurrent signals reported here could also reflect attentional modulations (*Hopf et al., 2006*; *Tsotsos et al., 1995*), this explanation is less likely. Rapid recurrent signals originating within the ventral pathway are involuntary (*Roland et al., 2006*) and temporally dissociable from top-down attentional signals that are typically reported at later times (*Boehler et al., 2008*; *Wyatte et al., 2014*).

Our study used a fixed button mapping for the yes/no responses on faces in the RSVP task, which in principle may introduce motor plan-related signals in the MEG responses. However we believe it is unlikely our results are confounded by motor-related signals for the following reasons: First, our experimental design used a delayed response design for the RSVP conditions (response was prompted 0.7–1 s after the offset of the stimulus), and no response for the 500 ms per picture condition. Thus any motor artifacts, including motor preparation signals, should have been delayed considerably for the RSVP conditions, and are completely absent in the 500 ms per picture condition. Second, we believe that our analysis to localize these processes with fMRI-MEG fusion supports the source of categorical information in IT. Third, previous studies have shown that motor-preparation signals occurs considerably later (300-400 ms after stimulus onset) (*Thorpe et al., 1996*). Thus it is unlikely the effects before 200 ms from stimulus onset reported in our data are related to motor preparation.

Taken together, the ultra-RSVP task enabled us to demarcate two different processing stages in the ventral stream, segregate the initial bottom-up sweep from late categorical signals, characterize the delay of the emergence of categorical information as an index of increased recurrent processing demands, and localize feedback activity in EVC. Our findings can motivate future experiments investigating the functional role of visual processes in the ventral pathway, capitalizing from the segregation of early and late processes achieved from the ultra-RSVP paradigm.

## Statiotemporal bounds for computational models of vision

Most popular computer vision models, such as deep neural networks (*LeCun et al., 2015*) and HMAX (*Riesenhuber and Poggio, 1999*), have adopted a feedforward architecture to sequentially transform visual signals into complex representations, akin to the human ventral stream. Recent work has shown that these models not only achieve accuracy in par with human performance in

many tasks (*He et al., 2015*), but also share a hierarchical correspondence with neural object representations (*Cichy et al., 2016b*; *Martin Cichy et al., 2017*; *Yamins et al., 2014*).

Even though models with purely feedforward architecture can easily recognize whole objects (*Serre et al., 2007*), they often mislabel objects in challenging conditions, such as incongruent object-background pairings, or ambiguous and partially occluded inputs (*Johnson and Olshausen, 2005*; *O'Reilly et al., 2013*). Instead, models that incorporate recurrent connections are robust to partially occluded objects (*O'Reilly et al., 2013*; *Rajaei et al., 2018*), suggesting the importance of recurrent processing for object recognition.

Unlike other studies that use stimuli that are occluded or camouflaged (*Spoerer et al., 2017*; *Tang and Kreiman, 2017*), our RSVP task offers no obvious computation that can be embued to feedback processes when presentation times are shortened. That is, our study does not inform on the precise nature of computations needed for stimulus evidence accumulation when presentation times are extremely short. Despite this fact, our findings on the duration and sequencing of ventral stream processes can still offer insights for developing computational models with recursive architecture. First, such models should solve object categorization within the first couple hundred milliseconds, even when the feedforward and feedback pathways are compromised as in the ultra-RSVP task. Second, the timing of recurrent processes should not be predetermined and fixed, but vary depending on viewing conditions, as in the case of onset and peak latency shifts in the RSVP decoding time series. Here viewing conditions related to the speed of the RSVP task, but it is reasonable to expect other challenging conditions, such as ambiguous or partial stimuli, to exert time delays, though possibly longer (*Tang and Kreiman, 2017*). Third, the timing and strength of the early visual signals should inversely determine the timing of categorical representations. And fourth, feedback processes in deep models should activate early layers of the model at the same time object representations are emerging at the last layers. Despite these insights, future research is needed to understand what recurrent computations are exactly carried out by the brain to solve visual recognition under RSVP-like presentation conditions.

## Ultra-fast rapid serial visual presentation as an experimental model to study recurrent neural processing

Since its conception (*Potter and Levy, 1969*), the RSVP paradigm has been implemented with stimulus rates about 100 ms per item or slower. Inherent to the design, RSVP experiments have revealed the temporal limitations of human perception (*Spence, 2002*), attention (*Nieuwenstein and Potter, 2006*), and memory (*Potter, 1993*). Recently however, behavioral investigations have been exploring even faster presentation rates in ultra-RSVP tasks (*Broers et al., 2018*; *Evans et al., 2011*; *Potter et al., 2014*). These experiments found that observers can detect target images at rates 13 to 20 ms per picture.

Due to its effectiveness in masking stimuli by combining forward and backward masking, the ultra-RSVP paradigm could be used to address the question whether recurrent processing is necessary for recognition of objects. One view posits that a purely feedforward mode of processing is sufficient to extract meaning from complex natural scenes (*DiCarlo et al., 2012*). High behavioral performance in the ultra-RSVP task has been used as an argument to support this view. Specifically, such rapid presentations of stimuli have been presumed to block recurrent activity, since low level visual representations are immediately overwritten by subsequent images and time is too short to allow multiple synaptic transmissions (*Tovée, 1994*).

However, this interpretation has been challenged both here, with the reengagement of EVC late in the processing stream, and by the finding that the ability to detect and categorize images at such speeds depends on the efficacy of the images to mask one another (*Maguire and Howe, 2016*). Thus, it still remains an open question whether recurrent activity is necessary to extract conceptual meaning (*Howe, 2017*).

Though our study did not address whether such recurrent activity can arise in more effective masking conditions that suppress visibility (*Maguire and Howe, 2016*), it paves the way for future studies to explore the link between stimulus visibility and recurrent neuronal processes. Such studies could vary the effectiveness of forward and backward masking to segregate the early from late visual signals, as accomplished here, and investigate under what conditions (e.g. ambiguous or occluded input) stimulus visibility (*King et al., 2016*; *Salti et al., 2015*) is associated with feedback activity. As ultra-RSVP reduces visibility, future studies could also investigate whether recurrent activity is an

integral component of the neural correlates of consciousness, defined as the minimum neuronal mechanisms jointly sufficient for a conscious percept (*Koch et al., 2016*).

## Materials and methods

### Participants

Seventeen healthy subjects (12 female; 16 right-handed and one left-handed; age mean ± s.d. 27.2 ± 5.7 years) with normal or corrected to normal vision participated in the RSVP experiment. They all signed an informed consent form and were compensated for their participation. The study was approved by the Institutional Review Board of the Massachusetts Institute of Technology and followed the principles of the Declaration of Helsinki.

### Experimental design and stimulus set

#### RSVP experiment

The stimulus set comprised 24 target images (12 faces and 12 objects) and 45 mask images of various object categories (*Figure 1a*). Images shown are not examples of the original stimulus set due to copyright; the exact stimulus set is visualized at https://megrsvp.github.io. We chose this stimulus set because it enabled comparison with MEG and fMRI data of a previous study (*Cichy et al., 2014*).

Participants viewed RSVP sequences of 11 images presented at a rate of 17 ms per picture or 34 ms per picture in separate trials. The middle image was randomly sampled from the set of 24 target images, and the surrounding images from the set of mask images. The stimuli were presented at the center of the screen against a gray background and subtended 2.9° of visual angle.

Each trial included a 0.5 s baseline time preceding the 17 ms per picture or 34 ms per picture RSVP sequence. At the end of the sequence a blank screen was presented for 0.7–1 s (uniformly distributed), which served to prevent motor artifacts, and then subjects were prompted to perform a two-alternative forced choice task reporting whether a face image was present in the sequence. That is, participants performed a yes/no task on faces without being informed on the alternative object (fruit/vegetable) target images. Responses were given with the right index finger using a MEG-compatible response box and a fixed button mapping for the face present and non-present response.

Trials were presented in random order in 12 blocks of 120 trials each, comprising both the 17 ms per picture and 34 ms per picture speed conditions interleaved randomly. In total, we collected 30 trials for each of the 24 target images and each of the two RSVP rates. The whole experiment lasted around 70 min. To avoid eye movement artifacts, the subjects were asked to fixate on a black cross presented at the center of the screen and blink only when pressing a button and not during the RSVP sequences.

Since the RSVP task was extremely challenging, before the experiment we trained participants for 5 min using a slower rate of 50 ms per picture. This assured the participants understood the task and could perform well during the higher presentation speeds.

#### 500 ms per picture experiment

In a separate MEG experiment, a different cohort of 16 healthy participants view the same 24 target images in isolation. Images were presented in random order for 500 ms with an ISI of 1 s. Participants were instructed to press a button and blink their eyes in response to a paper clip that was shown randomly every 3 to 5 trials. Participants completed 10–15 runs, with each image presented twice per run. We thus collected a total of 20–30 trials per target image. Note that the MEG data acquired for the 500 ms per picture experiment were part of a larger 92-image data set that has been previously published (*Cichy et al., 2014*).

#### fMRI experiment

The same 24 target images were also presented to a different cohort of 15 healthy participants in an fMRI experiment. In particular, images were presented in random order for 500 ms with an ISI of 2.5 s. Participants completed 2 sessions of 10 to 14 runs each, and each image was presented once per run. Thirty null trials with no stimulus presentation were randomly interspersed, during which the fixation cross turned darker for 100 ms and participants reported the change with a button press. This

resulted in 20–28 trials per target image. The fMRI data set was also part of a larger 92-image data set that has been previously published (*Cichy et al., 2014*).

## MEG acquisition and preprocessing

We collected MEG data using a 306-channel Elekta Triux system with a 1000 Hz sampling rate. The data was band-pass filtered with cut-off frequencies of 0.03 and 330 Hz. The MEG system contained 102 triple sensor elements (2 gradiometers and one magnetometer each) organized on a helmet shaped array. The location of the head was measured continuously during MEG recording by activating a set of 5 head position indicator coils placed over the head.

The raw MEG data was preprocessed with the Maxfilter software (Elekta, Stockholm) to compensate for head movements and denoise the data using spatiotemporal filters (*Taulu and Simola, 2006*; *Taulu et al., 2004*). The Brainstorm software (*Tadel et al., 2011*) was then used to extract trials from −300 ms to 900 ms with respect to target onset. Every trial was baseline-corrected to remove the mean from each channel during the baseline period, defined as the time before the onset of the first mask stimulus for the RSVP task, or the target image for the 500 ms per picture condition. A 6000 fT peak-to-peak rejection threshold was set to discard bad trials, and the remaining trials were smoothed with a 20 Hz low-pass filter. Eye blink artifacts were automatically detected from frontal sensor MEG data, and then principal component analysis was used to remove these artifacts.

## MEG multivariate pattern analysis

### Sensor space

To extract information about visual stimuli from the MEG data, we used multivariate pattern analysis. The procedure was based on linear support vector machines (SVM) using the libsvm software implementation (*Chang and Lin, 2011*) with a fixed regularization parameter C = 1. Before classification, the MEG trials for each target image were sub-averaged in groups of 3 with random assignment to reduce computational load, yielding $M$ trials per target image ($M$ was about 9–10 for the RVSP experiment when considering bad trials, and varied between 6 to 10 per subject for the 500 ms per picture condition).

The SVM analysis was performed separately for each subject in a time-resolved manner. Specifically, for each time point t (from −300 ms to 900 ms in 1 ms steps) the MEG sensor data were arranged in 306-dimensional pattern vectors, yielding $M$ pattern vectors per time point and target image. Then for each time point and pair of images separately, we measured the performance of the classifier to discriminate between images using leave-one-out cross-validation: $M$-1 vectors were randomly assigned to the training set, and the left-out vector to the training set to evaluate the classifier decoding accuracy. By repeating the classification procedure 100 times, each with random trial sub-averaging, and averaging decoding accuracies over repetitions, we populated a time-resolved 24 × 24 decoding matrix, indexed in rows and columns by the classified target images (*Figure 2a*). This decoding matrix, termed representational dissimilarity matrix (RDM), is symmetric and has an undefined diagonal (no classification within image). The entire procedure created one MEG RDM for each time point and subject.

Categorical division time series were constructed by dividing the MEG RDM matrix into partitions corresponding to within-category (face or object) and between-category stimulus comparisons. The difference of between-category minus within-category average decoding accuracies served as a measure of clustering by category membership. An alternate approach to computing categorical information time series is to directly train a classifier to discriminate face vs. object stimuli. While such methodological approach may be sensitive to different aspects of categorical stimulus information in general, it yielded consistent results in our data (*Figure 3—figure supplement 1*).

### Source space

To perform multivariate pattern analysis on the cortex and localize representational information on regions of interest (ROIs), we mapped MEG signals on source space. Source activation maps were computed on cortical surfaces derived from Freesurfer automatic segmentation (*Fischl et al., 2004*) of the Colin27 default anatomy (*Holmes et al., 1998*). The forward model was calculated using an overlapping spheres model (*Huang et al., 1999*). MEG signals were then mapped on a grid

of ~15000 cortical sources using a dynamic statistical parametric mapping approach (dSPM) (*Dale et al., 2000*) and time series were derived from sources within early visual cortex (EVC) and inferior temporal cortex (IT) (*Desikan et al., 2006*).

Source-based multivariate pattern analysis for the two cortical ROIs, EVC and IT, was performed exactly as in sensor space, however time-resolved pattern vectors were created by concatenating activation values from cortical sources within a given ROI, rather than concatenating the whole-head sensor measurements. This procedure resulted in one MEG RDM for each time point, ROI, and subject.

## Multidimensional scaling

To visualize the complex patterns of the 24 × 24 MEG RDMs, which capture the relations across the neural patterns elicited by the 24 target images, we used the first two dimensions of multidimensional scaling (MDS) (*Kruskal and Wish, 1978*; *Shepard, 1980*). MDS is an unsupervised method to visualize the level of similarity between different images contained in a distance matrix. Intuitively, MDS plotted the data in two dimensions where similar images were grouped together and different images far apart. fMRI data acquisition and analysis

The fMRI data was collected using a 3T Trio Siemens Scanner and 32-channel head coil. The structural images were acquired in the beginning of each session using T1-weighted sequences with TR = 1900 ms, TE = 2.52 ms, flip angle = 9°, FOV = 256 mm², and 192 sagittal slices. Functional data was acquired with high spatial resolution but partial coverage of the brain covering occipital and temporal lobe using gradient-echo EPI sequence with TR = 2000 ms, TE = 31 ms, flip angle = 80°, FOV read = 192 mm, FOV phase = 100%, ascending acquisition, gap = 10%, resolution = 2 mm isotropic, and slices = 25.

The details of fMRI analysis can be found in (*Cichy et al., 2014*) and here we explain it briefly. SPM8 (http://www.fil.ion.ucl.ac.uk/spm/) was used to analyze the fMRI data. The data was realigned, re-sliced, and co-registered with the structural images for each subject and session separately. Then a general linear model analysis was used to estimate t-value maps for each of the 24 target images. We further defined two volumetric ROIs for fMRI data analysis, V1 and IT. V1 was defined separately for each participant using an anatomical eccentricity template (*Benson et al., 2012*), and corresponded to a 0–6° visual angle. IT was defined using a mask comprising bilateral fusiform and inferior temporal cortex (*Maldjian et al., 2003*), keeping the most strongly 361 activated voxels from a cross-validated dataset to match the size of IT to the average size of V1.

## fMRI multivariate pattern analysis

To assess the relations between brain fMRI responses across the 24 target images, we constructed space-resolved fMRI RDMs using a correlation-based method. We conducted two types of analyses: (1) ROI-based and (2) spatially unbiased using a searchlight approach.

For the ROI-based analysis, we extracted and concatenated the V1 or IT voxel t-values to form ROI-specific fMRI pattern vectors. For each pair of images, we then calculated the dissimilarity (one minus Pearson's rho) between the fMRI pattern vectors, resulting in a 24 × 24 fMRI RDM indexed by the compared images. This procedure resulted in one fMRI RDM for each ROI and subject.

For the searchlight-based analysis (*Kriegeskorte et al., 2006*), we constructed fMRI RDMs for each voxel in the brain. In particular, for each voxel v we extracted fMRI activation values in a sphere centered at v with a radius of 4 voxels (searchlight at v) and arranged them into fMRI pattern vectors. For each pair of images, we then calculated the pairwise dissimilarity (one minus Pearson's rho) between fMRI pattern vectors, resulting in a 24 × 24 fMRI RDM indexed by the compared images. This procedure yielded one fMRI RDM for each voxel in the brain and subject.

## fMRI-MEG fusion using representational similarity analysis

To assess the spatiotemporal dynamics of EVC and IT, we applied a fMRI-MEG fusion approach based on representational similarity analysis (RSA) (*Kriegeskorte et al., 2008*; *Cichy et al., 2014*). The basic idea is that if two stimuli are similarly represented in MEG patterns, they should also be similarly represented in fMRI patterns, a correspondence that can be directly evaluated using the RDMs. Thus, we computed the similarity (Spearman's rho) between time-resolved MEG RDMs and space-resolved fMRI RDMs.

For ROI-based fMRI-MEG fusion, we used fMRI RDMs and MEG RDMs from the corresponding ROIs. In particular, for each time point we computed the similarity (Spearman's rho) between the subject-averaged MEG RDM and the subject-specific fMRI RDM. This procedure yielded $n = 14$ time courses of MEG-fMRI representational similarity for each ROI and subject.

### Temporal generalization of multivariate pattern analysis

To investigate whether maintenance of stimulus information was compromised in the RSVP relative to the 500 ms per picture condition, we extended the SVM classification procedure using a temporal generalization approach (*Cichy et al., 2014*; *Isik et al., 2014*; *King and Dehaene, 2014*; *Pantazis et al., 2017*). This method involved training the SVM classifier at a given time point $t$, as before, but testing across all other time points. Intuitively, if representations are stable over time, the classifier should successfully discriminate signals not only at the trained time $t$, but also over extended periods of time that share the same neuronal representations. We repeated this temporal generalization analysis for every pair of stimuli, and the results were averaged across compared images and subjects, yielding 2-dimensional temporal generalization matrices with the x-axis denoting training time and the y-axis testing time.

### Peak latency analysis

For statistical assessment of peak and onset latency of the time series, we performed bootstrap tests. The subject-specific time series were bootstrapped 1000 times and the empirical distribution of the peak latency of the subject-averaged time series was used to define 95% confidence intervals. A similar procedure was used to define 95% confidence intervals for onset latency. For peak-to-peak latency differences, we obtained 1000 bootstrapped samples of the difference between the two peaks, which resulted in an empirical distribution of peak-to-peak latency differences. We then used the tail of this empirical distribution to evaluate the number of bootstrap samples that crossed 0, which allowed us to compute a p-value for the peak-to-peak latency difference. Finally, the p-values were corrected for multiple comparisons using false discovery rate at a 0.05 level. A similar procedure was used for onset-to-onset differences.

We had one cohort of subjects for the RSVP conditions, and another for the 500 ms per picture condition. For consistency, we performed between-subject comparisons in all comparisons across the 17, 34, and 500 ms per picture conditions.

### Statistical inference

Statistical inference relied on non-parametric statistical tests that do not make assumptions on the distributions of the data (*Maris and Oostenveld, 2007*; *Pantazis et al., 2005*). Specifically, for the statistical assessment of classification time series, temporal generalization matrices, and MEG-fMRI representational similarities we performed permutation-based cluster-size inference. The null hypothesis was equal to 50% chance level for decoding results, and 0 for decoding differences or correlation values. In all cases we could permute the condition labels of the MEG data, which was equivalent to a sign permutation test that randomly multiplied subject responses by +1 or −1. We used 1000 permutations, 0.05 cluster defining threshold and 0.05 cluster threshold for time series and temporal generalization maps.

### Source data and code

Data and Matlab code used for statistical analyses and producing results in main *Figures 2*, *3* and *5*, are available as a Source data one file with this article.

## Acknowledgements

We are thankful to Aude Oliva and Molly Potter for discussions on the interpretations of results. This work was funded by the McGovern Institute Neurotechnology Program to DP and an Emmy Noether Award (CI241/1-1) to RMC. Data collection was conducted at the Athinoula A Martinos Imaging Center at the McGovern Institute for Brain Research, MIT.

## Additional information

### Funding

| Funder | Grant reference number | Author |
|---|---|---|
| McGovern Institute | Neurotechnology Program | Dimitrios Pantazis |
| Emmy Noether Award | CI241/1-1 | Radoslaw M Cichy |

The funders had no role in study design, data collection and interpretation, or the decision to submit the work for publication.

### Author contributions

Yalda Mohsenzadeh, Wrote the paper. Assisted with designing the experiments, Conducting the experiments, Analyzing the data, Preparing the manuscript; Sheng Qin, Assisted with designing the experiments, Conducting the experiments, Analyzing the data; Radoslaw M Cichy, Assisted with designing the experiments, And preparing the manuscript; Dimitrios Pantazis, Designed the experiment. Wrote the paper. Assisted with conducting the experiments, Analyzing the data, Preparing the manuscript

### Author ORCIDs

Yalda Mohsenzadeh (iD) https://orcid.org/0000-0001-8525-957X
Dimitrios Pantazis (iD) http://orcid.org/0000-0001-8246-8878

### Ethics

Human subjects: The study was approved by the Institutional Review Board of the Massachusetts Institute of Technology and followed the principles of the Declaration of Helsinki. All subjects signed an informed consent form and were compensated for their participation.

### Decision letter and Author response

Decision letter https://doi.org/10.7554/eLife.36329.015
Author response https://doi.org/10.7554/eLife.36329.016

## Additional files

### Supplementary files

• Source data 1. *Figures 2*, *3* and *5* source data and code. Decoding target images; resolving categorical information; and computing MEG-fMRI representational similarities.
DOI: https://doi.org/10.7554/eLife.36329.011

• Transparent reporting form
DOI: https://doi.org/10.7554/eLife.36329.012

### Data availability

All data generated or analysed during this study to support the main findings are included in the manuscript and supporting files. Source data files have been provided for Figures 2, 3 and 5.

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
