## [Decision Letter]

Thank you for submitting your article "Ultra-rapid serial visual presentation disentangles feedforward and feedback processes in the ventral visual pathway" for consideration by *eLife*. Your article has been reviewed by three peer reviewers, including Floris P de Lange as the Reviewing Editor and Reviewer #1, and the evaluation has been overseen by Michael Frank as the Senior Editor. The following individual involved in review of your submission has agreed to reveal their identity: Johannes Jacobus Fahrenfort (Reviewer #2).

The reviewers have discussed the reviews with one another and the Reviewing Editor has drafted this decision to help you prepare a revised submission.

Summary:

The goal of this study is to try to dissociate between feedforward and feedback processes during visual recognition. Human observers were engaged in a face vs. object categorization task using an RSVP paradigm. MEG decoding for all pairs of target images was attempted as a function of time. This lead to a first estimate of information flow interpreted as a feedforward sweep. Decoding was then attempted for face vs. object categorization and the associated information flow was delayed in time and interpreted as feedback processes.

The reviewers found this to be an original study using sound analyses and the main conclusions of the paper are well supported by the data. That said, there were concerns with respect to the analysis choice: why not directly try to classify faces vs. objects and instead perform a binary image classification followed by an inter- vs. intra-class averaging? There were different viewpoints on this issue by the different reviewers, so it would be required that you clarify the rationale for the analysis choice, and/or include the more direct inter-class classification analysis. Furthermore, there were concerns with respect to the relevance of the work for the development of computational models, and the interpretation of the results.

Essential revisions:

1) The analysis focuses on decoding image identity for pairs of images (either averaged over all pairs or looking at the difference when the pairs fall on the same side or on opposite side of the categorization boundary.) This is different from the task that the participants are engaged in (face vs. objects). Why did the authors not directly try to classify faces vs. objects, i.e. attempt to match the readout with the behavioral task? More generally, it is not completely clear how the various signals that are read out from MEG relate to behavioral decisions.

2) It is not clear how the various readouts that are performed relate to task-relevant neural computations. The authors discuss the relevance of their results for the development of computational models but this is not all that convincing because 1) there is no obvious computation that can be embued to the feedback when presentation times are shortened (unlike in some of the studies that are cited which use stimuli that are occluded or camouflaged); 2) the present study does not contribute significantly new knowledge that is directly relevant to constraining computational models (except for the fact that feedback should follow the feedforward pass in a way that is not "predetermined or fixed").

3) Figure 2E: Looks at peak latency to discuss limiting evidence accumulation. Eyeballing, the *onset* latencies in this graph are reliably faster for longer duration presentations (also consistently between within-face and within-object). I have observed similar effects in my own data (i.e. faster onsets for longer stimulus durations). This is potentially interesting to look at and discuss, because this not only speaks (1) to the emergence of feedforward activity but in addition (2) shows its early interaction with recurrent activity. It also goes against the argument that early decoding delinates feedforward activity alone, and paints a picture that is slightly more subtle. If early decoding would only reflect feedforward, we would not expect to see latency onset differences, but we do. The fact that different stimulus durations have different onsets suggests that feedback is already incorporated when the first decoding onsets emerge, going against the oversimplification that the early part of the decoding time course can be uniquely tied to feedforward alone. I think the manuscript would benefit from making this intricacy explict, the reader should not be left with the impression that we can fully separate feedforward and feedback. Figure 4A also suggests that feedback is already incorporated at a very early stage during long lasting presentations (recurrence mediated temporal generalization), and so do the onset latencies in Table 2.

4) "How did the disruption of the first sweep of visual activity, reported in the previous section, affect the emergence of categorical information in the RSVP conditions?" and "According to these theories, presenting stimuli at rapid presentation rates would compromise feedforward signals thus increase the need for recurrent processes to fill-in missing stimulus information."

Consider rephrasing this paragraph to more exactly reflect what is going on. Two things occur in the current paradigm / analyses:

1) From Figure 2: Short SOAs have less time to continue initial evidence accumulation (a process that in all likelihood already incorporates some local feedback, see previous comment).

2) From Figure 3 (and also according to the referenced theories) feedback signals are interrupted/hampered by the incoming feedforward signals of the mask at short SOAs (i.e. the feedforward signal of the ensuing mask interrupts the feedback signal of the target stimulus, see also Fahrenfort et al., 2007 and references below). This apparently slows down the speed and extent with which category information can be resolved using recurrence.

These two things could be better described/disentangled in this paragraph.

The current phrasing suggest that feedforward activation (rather than recurrence) is interrupted. Moreover, the current phrasing makes it sound as information is actually missing, as in Muckli et al., et al.2015.

This is not the case, there is not enough time to accumulate information in time, hence the contribution of recurrent processing to resolve image category.

Other relevant references in this context: Kovacs, Vogels and Orban, 1995;.O'Reilly et al., 2013; Fahrenfort et al., 2012.

5) Comparison of different conditions:

One set of participants engaged in the RSVP task (17 ms and 34 ms). A different set of participants engaged in the 1500 ms condition (recycled data from a previous experiment). This makes me wonder how the data are analyzed: in a within-subject, or between-subject fashion? What was the task for the 1500 ms condition? This should be added to the Materials and methods section.

6) Response mapping:

Was there a fixed button mapping, i.e. did subjects always press a particular button for a particular choice? If so, can the authors rule out that the 'categorical information' classified in Figure 3 pertains to the motor plan, rather than a perceptual categorization? Presumably the motor plan is also delayed for the more difficult conditions.

7) Peak latency analysis:

I didn't find any detail on how peak latency was established. Were they determined within every subject, or on the grand average (using jackknifing or boot-strapping to obtain the confidence intervals)? What was the exact procedure used to identify the peak? Given that most results hinge on the latency analyses, it is imperative that they are described in detail.

---

## [Author Response]

[…] The reviewers found this to be an original study using sound analyses and the main conclusions of the paper are well supported by the data. That said, there were concerns with respect to the analysis choice: why not directly try to classify faces vs. objects and instead perform a binary image classification followed by an inter- vs. intra-class averaging? There were different viewpoints on this issue by the different reviewers, so it would be required that you clarify the rationale for the analysis choice, and/or include the more direct inter-class classification analysis. Furthermore, there were concerns with respect to the relevance of the work for the development of computational models, and the interpretation of the results.

We thank the reviewers for highlighting these two critical points: a) the methodological choice of binary image classification followed by inter- vs. intra-class averaging; and b) the relevance of the work for the development of computational models.

Regarding point (a) – classifier choice:

We acknowledge that the two different methodological approaches – i) direct classification of faces vs. objects, and ii) binary image classification followed by inter- vs. intra-class averaging – may be sensitive to different aspects of categorical information in principle. However, both are valid and will often yield convergent results. Following the reviewers’ request, we performed direct categorical decoding on the data and the result is presented in Figure 3—figure supplement 1. The results, now added as supplemental material, largely replicate the results in Figure 3A (which uses binary classification followed by inter- vs. intra-class averaging).

The rationale behind using a binary classification in the main text is that we are also interested in comparing data from two different modalities (MEG and fMRI) using representational similarity analysis. Binary classification here creates the entries of RDM matrices from MEG data by detailing pair-wise classification results. We eventually compare the MEG RDMs and fMRI RDMs in Figure 5. Furthermore, in the context of this study the pairwise decoding approach is a more versatile approach, enabling us to study in one go within category decoding as well as categorical effect (between – within contrasts), and also construct multidimensional scaling visualizations as in Figure 3.

We now clarify our methodological choice to use binary classification MEG RDM matrices to construct categorical division time series in the Materials and methods section:

“Categorical division time series were constructed by dividing the MEG RDM matrix into partitions corresponding to within-category (face or object) and between-category stimulus comparisons. […] While such methodological approach may be sensitive to different aspects of categorical stimulus information in general, it yielded consistent results in our data (Figure 3—figure supplement 1).”

Regarding point (b) – relevance to computer models of vision:

We agree with the reviewers that there are limitations in how our work may contribute to the development of computational models. In particular, we acknowledge that unlike occlusion or camouflaged stimuli, shortening the presentation time of stimuli as in our RSVP task does not directly suggest an obvious computation for feedback processes. However, we do believe our work can still offer valuable insights in computational models with recurrent architectures. We have thus modified the Discussion to tone down our claims and clarify the above limitations.

“Unlike other studies that use stimuli that are occluded or camouflaged (Spoerer et al., 2017; Tang and Kreiman, 2017), our RSVP task offers no obvious computation that can be embued to feedback processes when presentation times are shortened. […] Despite these insights, future research is needed to understand what recurrent computations are exactly carried out by the brain to solve visual recognition under RSVP-like presentation conditions.”

Essential revisions:1) The analysis focuses on decoding image identity for pairs of images (either averaged over all pairs or looking at the difference when the pairs fall on the same side or on opposite side of the categorization boundary.) This is different from the task that the participants are engaged in (face vs. objects). Why did the authors not directly try to classify faces vs. objects, i.e. attempt to match the readout with the behavioral task? More generally, it is not completely clear how the various signals that are read out from MEG relate to behavioral decisions.

We thank the reviewers for highlighting this point. Please see our detailed response to the summary above, which clarifies the merits of performing binary classification (i.e. allows us to i) compare data from two different modalities – MEG and fMRI; ii) perform multidimensional scaling; and iii) characterize both intra- and inter-class decoding). We also demonstrate that this method yields largely equivalent results to direct classification of faces vs. objects. These results are now added as Figure 3—figure supplement 1.

Regarding the behavioral data, since this was a delayed response task, only accuracy and not reaction times are relevant. Performance on the 34ms per picture condition was almost perfect (i.e. nearly all trials had correct ‘seen’ responses), and thus only the 17ms per picture condition, with performance consistently above chance (Figure 1C), is interesting for behavioral analysis. But unfortunately, in our paradigm correct and incorrect ‘seen’ trials greatly varied across stimuli and participants, and often individual stimuli were consistently seen or unseen. This confounds any analysis of behavioral decisions with individual stimuli. Future work conducting experiments which adjust visibility per stimulus and participant are needed to better clarify the relation of behavior to neural signals. Nevertheless, following the reviewers’ comment, we performed an analysis to decode seen versus unseen *face* trials in the 17ms per picture RSVP condition (presented in Figure R1). The results show we can decode the seen vs. unseen condition (subject to the confound above). Such analysis is not possible for the 34ms per picture condition because nearly all trials had correct ‘seen’ responses. Since this is not the focus of the current study, and given these results are confounded by stimulus identity, we opted not to include these results in the manuscript.

**Author response image 1. respfig1:** Decoding seen versus unseen faces in the 17ms per picture condition.

2) It is not clear how the various readouts that are performed relate to task-relevant neural computations. The authors discuss the relevance of their results for the development of computational models but this is not all that convincing because 1) there is no obvious computation that can be embued to the feedback when presentation times are shortened (unlike in some of the studies that are cited which use stimuli that are occluded or camouflaged); 2) the present study does not contribute significantly new knowledge that is directly relevant to constraining computational models (except for the fact that feedback should follow the feedforward pass in a way that is not "predetermined or fixed").

We agree with the reviewer that, unlike occlusion or camouflaged stimuli, shortening the presentation time of stimuli as in our RSVP task does not offer an obvious computation for feedback processes. We thus acknowledge the limitations of our work in informing the design of computational models. However we do believe our work does offer valuable insight for computational models with recurrent architectures. For example, we believe the onset and peak latency shifts suggest explicit temporal constraints for the representational dynamics of recurrent networks; and the inverse timing of early signals with respect to late categorical representations suggest biologically plausible interactions that support object recognition in computation networks. We have thus modified the Discussion to tone down our claims, clarify the above limitations, and better specify how this work can contribute to the design of computational models.

“Unlike other studies that use stimuli that are occluded or camouflaged (Spoerer et al., 2017; Tang and Kreiman, 2017), our RSVP task offers no obvious computation that can be embued to feedback processes when presentation times are shortened. […] Despite these insights, future research is needed to understand what recurrent computations are necessary to solve visual recognition under shortened stimulus presentations related to our RSVP task.”

3) Figure 2E: Looks at peak latency to discuss limiting evidence accumulation. Eyeballing, the onset latencies in this graph are reliably faster for longer duration presentations (also consistently between within-face and within-object). I have observed similar effects in my own data (i.e. faster onsets for longer stimulus durations). This is potentially interesting to look at and discuss, because this not only speaks (1) to the emergence of feedforward activity but in addition (2) shows its early interaction with recurrent activity. It also goes against the argument that early decoding delinates feedforward activity alone, and paints a picture that is slightly more subtle. If early decoding would only reflect feedforward, we would not expect to see latency onset differences, but we do. The fact that different stimulus durations have different onsets suggests that feedback is already incorporated when the first decoding onsets emerge, going against the oversimplification that the early part of the decoding time course can be uniquely tied to feedforward alone. I think the manuscript would benefit from making this intricacy explict, the reader should not be left with the impression that we can fully separate feedforward and feedback. Figure 4A also suggests that feedback is already incorporated at a very early stage during long lasting presentations (recurrence mediated temporal generalization), and so do the onset latencies in Table 2.

We thank the reviewer for directing our attention to onset latencies. The reviewer offers great insights in the role of variable onset latencies as an index of rapid recurrent activity. Convinced by the reviewer’s argument, we modified Figure 2H, I, J and the Results and Discussion sections to include these results.

The reviewer mentioned has observed similar results in his/her data; we were unable to find a related publication, but we are of course happy to cite this work if the reviewer shares the related information.

“Fourth, onset latencies shifter later with faster stimulus presentation rates (Figure 2H). That is, the 0.5 s per picture condition had onset at 28 ms (9-53 ms), followed by the 34 ms per picture RSVP condition at 64 ms (58-69 ms), and finally the 17 ms per picture RSVP condition at 70 ms (65-77 ms) (all statistically different; *P*< 0.05; two-sided sign permutation tests).”

“The decreased decoding accuracy combined with the increasingly early peak latency and increasingly late onset latency for the RSVP conditions indicate that visual activity was disrupted over the first 100 ms. […] The fact that different stimulus durations have different onsets suggests that interactions with recurrent activity are already incorporated when the first decoding onsets emerge, arguing against the view that the early part of the decoding time course can be uniquely tied to feedforward alone (Fahrenfort et al., 2012; Lamme and Roelfsema, 2000; Ringach et al., 1997).

“In sum, decoding accuracies decreased with progressively shorter stimulus presentation times, indicating that neuronal signals encoded less stimulus information at rapid presentation rates. […] Importantly, the progressively earlier peak with shorter presentation times indicated disruption of the first sweep of visual activity, thus indexing feedforward and local recurrent processing and segregating it in time from subsequent processing that includes feedback influences from high-level visual cortex.”

“Reduced early recurrent activity in the RSVP conditions could be due to two reasons. […] This may be reflected in the progressively later onset latencies with decreasing viewing times in Figure 2H, I, J which suggest that local recurrent interactions have a rapid influence in the ventral stream dynamics.”

4) "How did the disruption of the first sweep of visual activity, reported in the previous section, affect the emergence of categorical information in the RSVP conditions?" and "According to these theories, presenting stimuli at rapid presentation rates would compromise feedforward signals thus increase the need for recurrent processes to fill-in missing stimulus information."Consider rephrasing this paragraph to more exactly reflect what is going on. Two things occur in the current paradigm / analyses:1) From Figure 2: Short SOAs have less time to continue initial evidence accumulation (a process that in all likelihood already incorporates some local feedback, see previous comment).2) From Figure 3 (and also according to the referenced theories) feedback signals are interrupted/hampered by the incoming feedforward signals of the mask at short SOAs (i.e. the feedforward signal of the ensuing mask interrupts the feedback signal of the target stimulus, see also Fahrenfort et al., 2007 and references below). This apparently slows down the speed and extent with which category information can be resolved using recurrence.These two things could be better described/disentangled in this paragraph.The current phrasing suggest that feedforward activation (rather than recurrence) is interrupted. Moreover, the current phrasing makes it sound as information is actually missing, as in Muckli et al., 2015.This is not the case, there is not enough time to accumulate information in time, hence the contribution of recurrent processing to resolve image category.Other relevant references in this context: Kovacs, Vogels and Orban, 1995; O'Reilly et al., 2013; Fahrenfort et al., 2012.

We agree with the reviewer and rephrased the paragraph as requested. Also, we now cite the above references in several places in the manuscript, including below.

“However, opposing theories concur that feedback activity is critical for visual awareness and consciousness (Lamme and Roelfsema,

2000; Ahissar et al., 2009; Fahrenfort et al., 2017, 2012). […] These would be consistent with slowing down the speed and extent with which category information can be resolved using recurrence

(Brincat and Connor, 2006; Tang and Kreiman, 2017).”

5) Comparison of different conditions:One set of participants engaged in the RSVP task (17 ms and 34 ms). A different set of participants engaged in the 1500 ms condition (recycled data from a previous experiment). This makes me wonder how the data are analyzed: in a within-subject, or between-subject fashion? What was the task for the 1500 ms condition? This should be added to the Materials and methods section.

Indeed, we have one cohort subjects for the RSVP conditions and another for the 500ms per picture condition (called 1500ms in the previous version of the manuscript). To be consistent, we performed between-subject comparisons in all cases. This is now clarified in the method section of the revised manuscript.

“We had one cohort of subjects for the RSVP conditions, and another for the 500 ms per picture condition. For consistency, we performed between-subject comparisons in all comparisons across the 17 ms, 34 ms, and 0.5 ms per picture conditions.”

For the 500 ms per picture condition, participants performed simple detection tasks to keep them engaged, which were adapted to the specific requirements of each acquisition technique. Specifically, for the MEG experiment, participants were instructed to press a button and blink their eyes in response to a paper clip that was shown randomly every 3 to 5 trials. For the fMRI experiment, thirty null trials with no stimulus presentation were randomly interspersed, during which the fixation cross turned darker for 100 ms and participants reported the change with a button press. The description of the task in this condition is now clarified in the Materials and methods section of the revised manuscript:

“Participants were instructed to press a button and blink their eyes in response to a paper clip that was shown randomly every 3 to 5 trials”

“Thirty null trials with no stimulus presentation were randomly interspersed, during which the fixation cross turned darker for 100 ms and participants reported the change with a button press.”

6) Response mapping:Was there a fixed button mapping, i.e. did subjects always press a particular button for a particular choice? If so, can the authors rule out that the 'categorical information' classified in Figure 3 pertains to the motor plan, rather than a perceptual categorization? Presumably the motor plan is also delayed for the more difficult conditions.

We used a fixed button mapping, and we revised the manuscript to clarify this point. We acknowledge the reviewer’s concern regarding the potential confounding nature of the motor plan signals. However we believe it is unlikely our results are motor-related for the following reasons: First, our experimental design uses a delayed response design for the RSVP conditions (response was prompted 0.7-1 s after the offset of the stimulus), and no response for the 500 ms per picture condition. Thus any motor artifacts, including motor preparation signals, should have been delayed considerably for the RSVP conditions, and are completely absent in the 500 ms per picture condition. Second, we believe that our analysis to localize these processes with fMRI-MEG fusion which shows again progressively later peak latencies in fMRIMEG fusion time series in IT supports the source of this categorical information and its pattern in IT. Third, previous studies, such as Thorpe et al., 1996, have shown that motor-preparation signals occurs considerably later (300ms-400ms after stimulus onset). Thus it is unlikely the effect reported in our data (which is before 200ms after stimulus onset) to be related to motor-preparation.

“Our study used a fixed button mapping for the yes/no responses on faces in the RSVP task, which in principle may introduce motor plan-related signals in the MEG responses. […] Thus it is unlikely the effects before 200ms from stimulus onset reported in our data are related to motor preparation.”

7) Peak latency analysis:I didn't find any detail on how peak latency was established. Were they determined within every subject, or on the grand average (using jackknifing or boot-strapping to obtain the confidence intervals)? What was the exact procedure used to identify the peak? Given that most results hinge on the latency analyses, it is imperative that they are described in detail.

Peak and onset latency 95% confidence intervals, and peak-to-peak and onset-toonset latency differences were evaluated by bootstrapping across participants and computing subject-averaged time series. Thus, the peaks and onsets were defined on (original or bootstrapped) subject-averaged time series.

The exact procedure was originally described in the ‘Statistical inference’ subsection of the Materials and methods section. We now moved the text to a separate section ‘Peak latency analysis’, which we revised for further clarification.

“Peak latency analysis

For statistical assessment of peak and onset latency of the time series, we performed bootstrap tests. […] We had one cohort subjects for the RSVP conditions, and another for the 0.5 ms per picture condition. To be consistent, we performed between-subject comparisons in all comparisons across the 17 ms,

34 ms, and 0.5 ms per picture conditions.”